# Naturally Occurring Endocrine Disorders in Non-Human Primates: A Comprehensive Review

**DOI:** 10.3390/ani12040407

**Published:** 2022-02-09

**Authors:** Jaco Bakker, Melissa A. de la Garza

**Affiliations:** 1Biomedical Primate Research Centre (BPRC), Animal Science Department (ASD), 2288GJ Rijswijk, The Netherlands; 2Independent Researcher, San Antonio, TX 78216, USA; mdelagarzadvm@gmail.com

**Keywords:** adrenal, endocrine disorders, hormone, neoplasia, NHP, pancreas, parathyroid, pituitary, reproductive organs, thymus, thyroid

## Abstract

**Simple Summary:**

Nonhuman primates (NHP) can become ill due to a variety of diseases and disorders, yet endocrine disorders remain underreported. Therefore, an exhaustive literature search on this subject via widely used academic search systems, peer-reviewed publications, proceedings, and newsletters was performed. Selected major endocrine entities will be described emphasizing clinical signs, morphologic features, concomitant diseases, as well as available treatment options. In most cases, no clinical signs were noted and on gross pathology, the endocrine organs were unremarkable. The diagnosis was frequently made as incidental findings after standard histological examination. Although the findings were frequently incidental many have the potential to impact studies. This review explains that there is no standard procedure for diagnosing, monitoring, or treating endocrine disorders in NHP. More research is needed to evaluate these procedures and establish risk factors.

**Abstract:**

Literature concerning veterinary medicine of non-human primates is continuously updated, yet endocrine disorders remain underreported. While case or survey reports of individual endocrinopathies are available, a comprehensive review is not. An exhaustive literature search on this subject via widely used academic search systems, (e.g., Google Scholar, PubMed, BioOne complete and Web of Science), and peer-reviewed publications, proceedings, and newsletters was performed. Selected major endocrine entities will be described with emphasis on clinical signs, morphologic appearances, concomitant diseases, as well as available treatment options. Mostly, no clinical signs were noted and on gross pathology, the endocrine organs were unremarkable. An endocrine-related diagnosis was frequently made as an incidental finding after standard histopathological examination. During the review, the pancreas represented the most affected endocrine organ and diabetes mellitus represented the most clinically significant disorder. Currently, no standard procedure for diagnosing, monitoring, or treating endocrine disorders in non-human primates exists.

## 1. Introduction

The increased presence of non-human primates (NHP) in zoos, sanctuaries, and primate centers (NHP as a biomedical research model) has required refinement of welfare and veterinary care and practices to facilitate optimal clinical care and research support. Like humans, NHP can become ill due to a variety of causes, including endocrine disorders. The endocrine system plays a vital role in regulating body functions through the release and uptake of hormones. However, while there is an abundance of information available on the spontaneous endocrine disorders of NHP, most information is fragmented, dated, narrow in focus, or limited to single case reports. An in-depth review of endocrine disorders in NHP is lacking [1]. To these ends, we first conducted a literature search for books, book chapters, peer-reviewed publications, conference proceedings, and newsletters in academic literature databases, such as Google Scholar, PubMed, BioOne Complete, and Web of Science, using words and word combinations, such as endocrine disease, thyroid, pancreas, diabetes, monkeys. We then evaluated the search results for those reports that we considered as clinically relevant and then divided them into the endocrine organs (Table 1). Additionally, historical publications are disappearing from libraries and are often not digitally available. Some publications are limited to simply online abstracts, making data verification impossible, and, therefore, will not be included in this review. Experimentally related disorders due to the creation of human research models or side effects of experimental manipulation are not discussed in this review. This thorough review of naturally occurring endocrine disorders in NHP provides a concise reference for veterinarians, investigators, and others engaged in the practice of managing NHP. The following glands of the endocrine system will be discussed: adrenal, hypothalamus, pancreas, parathyroid, pineal, pituitary, reproductive organs (ovaries and testes), thymus, and thyroid (Table 1).

## 2. Hypothalamus

No disorders of NHP involving the hypothalamus have been reported. However, Goncharova and Lapin [90,91] found that in baboons and macaques, age has a prominent effect on the hypothalamic–pituitary–adrenal axis. Basal plasma levels of adrenal androgenes and early precursors of steroid hormones progressively decrease with age, while cortisol concentrations do not change.

## 3. Pituitary Gland

Reported dysfunctions of the pituitary gland are mostly related to the presence of tumors and are primarily reported in macaques. The overwhelming majority are benign adenomas with concomitant hormone dysregulation, such as pheochromocytomas. Some tumors may be of both secretory and non-secretory cell origin, others may be composed of multiple tumor types and in some cases, tumors may involve multiple endocrine organs. Tumors of the pituitary gland can cause pathology by increased secretion of hormones, or as space-occupying masses.

Chalifoux et al. [2] and Beniashvili [3] described one pituitary tumor in a rhesus monkey (*Macaca mulatta*) structurally resembling a chromophobe adenoma which caused various hormonal disorders. A mass was found in the pituitary gland that was identified as a chromophobe adenoma consisting primarily of sparsely granulated cells. Other lesions of note included gynecomastia, galactorrhea, testicular atrophy, ankylosing spondylitis, and amyloid present in the adrenal gland, intestinal tract, liver, and spleen.

Tsuchitani and Narama [4] reported a case of an adult male cynomolgus monkey (*Macaca fascicularis*) with no significant clinical signs. On necropsy, an enlarged pituitary gland was found (6 mm in diameter) and consisted almost entirely of a neoplastic mass. The thyroid gland was twice its normal size and diagnosed as a colloid goiter. Based on gross and histological findings, classifying this tumor as chromophobe adenoma derived from pars distalis of adenohypophysis.

A case of corticotropic cell pituitary adenoma in an adult male golden lion tamarin (*Leontopithecus rosalia*) is reported by Dias et al. [5]. The animal was found weak and recumbent without prior signs of illness. On histopathology, a 1 mm circumscribed pituitary mass was observed. The authors continue describing a case of an adrenal pheochromocytoma and corticotropic cell pituitary adenoma in a geriatric male brown spider monkey. The pituitary gland was enlarged and comprised well-differentiated cuboidal to columnar, granular cells.

A case of an adult male cynomolgus monkey presented with cervical ulcerative dermatitis and bilateral galactorrhea was described by Daviau and Trupkiewicz [6]. Lab work revealed persistent hyperglycemia and elevated serum prolactin and cortisol levels. A prolactin-secreting pituitary tumor was diagnosed antemortem.

Cianciolo and Hubbard [7] reported one pituitary carcinoma in a baboon.

In 2006, Remick et al. [8] characterized pituitary adenomas in a colony of cynomolgus monkeys. Fourteen out of 491 animals were affected (2.9%) and in total 35 adenomas were identified. Several tumors presented with major enlargement of the pituitary gland while others were identified histologically as multifocal microadenomas. The most prevalent type were prolactin-secreting neoplasms. These data indicate pituitary adenomas in cynomolgus monkeys often appear to have mixed histologic outcomes and hormone expression.

Guzman and Radi [13] described pituitary hyperplasia/hypertrophy in a female cynomolgus monkey that presented as chronic lymphocytic thyroiditis and thyroid acinar atrophy, causing a breakdown of the negative feedback loop and subsequent dysregulation of hormone secretion.

Guardado-Mendoza et al. [9] found numerous cases of adenomas in baboons, more common in females.

Confer et al. [10] described a multiple endocrine neoplasia-like syndrome in baboons where up to four endocrine organs were affected. The majority of these baboons had adenoma or hyperplasia involving the pituitary gland.

Simmons and Mattisson [11] reported adenomas in five and a chromophobe adenoma in one rhesus monkey.

Stoneburg et al. [12] reported a case of female golden lion tamarin being presented with progressive, bilaterally symmetric alopecia and weight loss. CT revealed a 0.5 cm mass arising from the pituitary fossa. Necropsy revealed a large pituitary mass, and histopathology of the mass was consistent with a pituitary pars intermedia adenoma. Bilateral adrenal cortices were hyperplastic consistent with Cushing’s disease, with cortical: medullary ratios of approximately 3:1 and expansion of primarily the zona fasciculata (see Section 8: adrenal glands).

## 4. Pineal Gland

To date, no disorders involving the pineal gland have been reported in NHP.

## 5. Parathyroid Gland

### 5.1. Hyperparathyroidism

In 1962, Jeffree reported hyperparathyroidism in three Humboldt’s woolly monkeys (*Lagothrix lagothricha*), one Capuchin, and one African green monkey (*Chlorocebus aethiops*) [14]. Four exhibited bone abnormalities, and all five had severely enlarged parathyroid glands. The author concluded diet likely played a role.

Dias et al. [5] reported a case of a 14-year-old male mantled howler monkey (*Alouatta palliata*) presented with signs of wasting and gastrointestinal distress. Major findings on necropsy included ulcerative gastroenterocolitis, severe interstitial nephritis, and glomerulonephritis. Histologically, a parathyroid adenoma, an adrenal gland pheochromocytoma, and a pancreatic islet cell adenoma were diagnosed.

In 1998, Hatt and Sainsbury [15] reported a metabolic bone disease in a subadult female common marmoset (*Callithrix jacchus*) with findings consistent with poor growth and wasting as well as severe bilateral nephrocalcinosis. Osteomalacie was most likely due to vitamin D3 and calcium deficiency and a presumptive diagnosis of nutritional secondary hyperparathyroidism.

A benign solitary primary hyperparathyroidism was diagnosed in one adult female olive baboon (*Papio anubis*) [16]. Aside from elevated serum calcium concentrations, no other abnormalities were noted.

In more recent literature, Guardado-Mendoza et al. [9] reported two cases of parathyroid hyperplasia in baboons.

### 5.2. Neoplasia

Confer et al. [10] and Guardado-Mendoza et al. [9] both reported an adenoma in the parathyroid gland of a baboon. Similarly, Simmons and Mattisson [11] reported an adenoma and carcinoma in rhesus monkeys.

## 6. Thymus

Kotani et al. [17] reported on a clinically and behaviorally normal 4-year-old male cynomolgus monkey with a 1.5 cm mass in the thymus found incidentally at necropsy. Due to the presence of spindle and cortex-like lymphoid cells, a mixed thymoma was diagnosed.

## 7. Pancreas

### 7.1. Diabetes mellitus

The most frequently reported, naturally occurring pancreatic disorder in NHP is by far diabetes mellitus (DM) [18,19,20,21,22,23,24,25,26,27,28,29,30,31,32,33,34,35,36,37,38]. Extensive research regarding the spontaneous development, characteristics, and comorbidities of DM in NHP has mainly been conducted in cynomolgus and rhesus monkeys [20,22,23,30,37,92]. Type 2 diabetes mellitus (T2DM) is prevalent in NHP than type 1 diabetes mellitus (T1DM). Gestational diabetes mellitus (GDM) [39,93] has been reported less in NHP, with complications such as those observed in human GDM. Other forms of diabetes also occur in NHP.

The majority of NHP DM cases are T2DM which is characterized by insulin resistance. First, T2DM is characterized by normal glucose tolerance followed by insulin resistance, resulting in a compensatory increase in insulin secretion, and deterioration of carbohydrate metabolism. Age and glucose values seem to be related to NHP [94,95]. T2DM is associated with aging [36,37,40]. Hemoglobin A1c (HbA1c) levels are associated with age in non-diabetic humans [96,97,98], unlike macaques and squirrel monkeys [99,100]. The development of T2DM is also associated with excessive adiposity or obesity. In overweight macaques, a long-term pre-diabetic state (for several years) is demarcated by increased insulin levels and enhanced beta-cell responsiveness. When this insulin surge begins to decline, hyperglycemia and overt T2DM are observed. Over time their glycemic profile deteriorates, and animals often lose weight. Obesity and insulin resistance could relate to a number of factors, e.g., genetic predisposition, overall poor dietary choices, stress, and contraception [21,23,37].

Clinically, NHP with DM exhibits polydipsia, polyuria, weight loss (late stage), polyphagia, and lethargy. Suggestive for the presence of DM is animals or conspecifics drinking the urine of the DM animal. Regardless of the type of diabetes, fasting serum hyperglycemia is the most common diagnostic used as an indicator of DM. Hyperglycemia in NHP is typically defined as blood glucose levels higher than 126 mg/dl (= 5.6 mmol/L) [37]. However, elevated blood glucose levels in anesthetized animals may be dismissed as an artifact of capture, restraint, or anesthetics as they are a symptom of DM. Therefore, in laboratory tests for the diagnosis of NHP suspected of having DM, a complete blood count (CBC), serum chemistry, levels of insulin, triglyceride, fructosamine, and HbA1c should be included. For macaques, normal fasting serum glucose levels, fasting serum glucose levels and fructosamine values are described by several authors [19,36,37,101,102,103,104,105,106]. Further, hypertriglyceridemia, hypercholesterolemia, glycosuria, and ketonuria can be present during illness. Obese, insulin-resistant nondiabetic, and T2DM cynomolgus monkeys were reported to exhibit aberrant lipid and lipoprotein metabolism, which included increased cholesterol, triglycerides, and free fatty acid levels and decreased HDL cholesterol [38]. Moreover, high levels of serum glycated proteins are found, such as HbA1c and fructosamine. Measuring serum fructosamine and HbA1c has numerous advantages. Whereas blood glucose measurement represents a snapshot in time and may transiently be elevated or decreased with food intake, fasting, stress, activity, or illness, fructosamine is unaffected by these factors, and the concentration represents an average blood glucose level for the two-to-three-week period preceding blood sampling [104]. HbA1c levels reflect blood glucose levels over the past 8–12 weeks [41]. In addition, HbA1c and fructosamine concentrations are unaffected by recent food intake, because surfactants have been added to the second-generation assay to overcome lipemia, thus eliminating the need for presampling fasting. Fructosamine and HbA1c are significantly correlated in diabetic and non-diabetic cynomolgus monkeys [19]. Furthermore, measurement of both fructosamine and glucose may result in early diagnosis of DM, thereby allowing earlier therapeutic intervention.

Although challenging, unsedated sampling of capillary blood for instant glucose measurements is preferred. Positive reinforcement training (PRT) may be used to train NHP to present a body part for a skin prick. Blood glucose levels may then be obtained with the use of a human glucometer. If the animal is uncooperative in presenting a body part for blood sampling, the animal can be squeezed gently, to obtain the sample.

Intravenous glucose tolerance testing (IVGGT) is most used as an early screening test of DM and to help monitor disease progression in NHP [107]. IVGTT is performed under anesthesia. As anesthetics can alter blood glucose levels, a standardized sedation protocol is necessary [108,109,110,111,112]. A T2DM animal should normally show persistently elevated plasma glucose and insulin levels in response to IVGTT [107]. Reduced insulin levels indicate T1DM. In addition, IVGTT is useful for diagnosing prediabetic animals [18,36,37,113].

Glucose in the urine as a diagnostic test for DM is challenging as the specific level at which glucose spills from the blood into the urine is not determined for NHP. In addition, individuals can be clinically diabetic and yet not pass glucose into the urine. Therefore, it is advisable that multiple diagnostic tests take place to diagnose DM, particularly in the early stages of DM. Additionally, obtaining representative urine samples to test for glycosuria can be problematic depending on husbandry and housing practices in NHP.

As in human diabetes management, making modifications to limit caloric uptake and stimulate physical and social activity (including sex) can help to prevent, reduce the symptoms, slow down disease progression, and control (pre)DM in NHP [38,114,115,116,117,118,119]. Adjusting a diet towards more natural compositions decreased obesity-related problems and improved animal welfare, e.g., increasing fiber and decreasing sugar content resulted in the reversal of prediabetes and more natural behavior in great apes [120]. However, strategies to individualize caloric intake and encourage exercise are difficult to implement in socially housed NHP [121]. Caretakers should be educated on dietary modifications and restrictions [30,122]. The regular primate pallets often contain high levels of carbohydrates, particularly problematic simple sugars. Additionally, the prevention of obesity can significantly decrease DM. T2DM is preceded by a long period of insulin resistance (often obesity related), and only requires exogenous insulin therapy if pancreatic islet reserves are depleted [36]. Long and short-acting insulin are available to tailor treatment to the individual animals’ needs. The goal of insulin treatment is to reduce hyperglycemia to normal ranges, which will improve quality of life by relieving symptoms and preventing the complications of DM [20,30,122,123,124,125]. However, it is challenging to monitor glucose levels and administer daily insulin in NHP. Additionally, behavioral and clinical observations are necessary to monitor the side effects of the treatment and detect any acute or chronic complications of T2DM as it is a progressive disease. As noted in humans, acute and chronic complications of DM include kidney disease, ocular disease, micro- and macroangiopathy, hyperosmolar and ketotic coma [126]. Weekly bodyweight measurement is advisable as it provides vital information on the animal’s clinical state.

T2DM has also been reported in aging captive chimpanzees (*Pan troglodytes*). Reference intervals for fasting plasma glucose and HbA1c show a positive correlation and revealed that the overall occurrence of T2DM in chimpanzees is nearly five times greater in aged individuals than in the general population [41]. Rosenblum et al. [42] reported noninsulin-dependent DM in four adult chimpanzees.

Jones et al. [40] reported DM in their colony of sooty mangabeys (*Cercocebus atys*). They concluded that T2DM is more common in mangabeys than in other NHP (overall prevalence of DM was 11% versus less than 6% reported for other NHP). Diabetes in mangabeys show some extraordinary clinicopathologic characteristics, e.g., absence of altered cholesterol levels and glycated hemoglobin however show a robust association of pancreatic insular amyloidosis with clinical DM.

Obese African green monkeys were reported to develop insulin resistance and dyslipidemia which progressed to T2DM [18]. Some were insulin sensitive with abundant islet insulin staining but were hyperglycemic. The authors suggest a strong heritable pattern, such as maturity-onset diabetes of the young or mitochondrial diabetes.

Obese marmosets were reported to be diabetic. They showed an increase in body fat with little change in lean mass, blood glucose, HbA1c levels, and triglycerides besides a very-low-density lipoprotein cholesterol [23].

GDM has been described in cynomolgus and rhesus monkeys during pregnancy [23,39,43]. Fasting blood glucose levels are decreased during gestation. However, in NHP, GDM is associated with elevated glucose and insulin levels, which may result in macrosomic offspring. During gestation, the disease is usually without clinical symptoms and may only be noted after the delivery of a larger than normal infant. In most cases of GDM, insulin treatment is not necessary, and the animal reverts to normoglycemia postpartum.

T1DM is a rare condition usually diagnosed in infant and sub-adult NHP. T1DM is primarily caused by the autoimmune destruction of pancreatic cells. Therefore, the pancreas cannot produce insulin. Treatment for T1DM is lifelong insulin therapy and diet modifications. Although labor and time consuming, a high quality of life is achievable with proper medical management.

### 7.2. Acute and Chronic Pancreatitis

The lines between acute and chronic disease are not always clear. Endocrine failure occurs due to progressive destruction of the gland due to chronic inflammation of the pancreas, which results in pancreatogenic or type 3c diabetes mellitus (T3cDM). In humans, T3cDM is frequently misdiagnosed as T2DM due to difficulties in differentiating between T2DM and T3cDM [44]. Therefore, the literature regarding T3cDM is likely lacking in NHP.

Chandler et al. [45] described a case of a juvenile rhesus monkey that died suddenly. Necrotizing pancreatitis associated with adenovirus was diagnosed.

McClure and Chandler [32] conducted a survey of pancreatic lesions in NHP showing 18.7% had lesions of varying severity. The most common lesion included focal areas of mononuclear cellular inflammation with varying degrees of periductal fibrosis.

Spontaneous pancreatitis has been reported in NHP [32,46,47] and at least two of these cases were caused by adenovirus infection [32,46].

Doepel et. al. [47] reported cases of chronic pancreatitis with a final acute assault of the pancreas as the causes of death to both a spider monkey (*Ateles* sp.) and squirrel monkey (*Saimiri* sp.) Both monkeys had a short history of illness prior to death.

Völker and Plesker [48] reported a case of T2DM diagnosed in an adult female cynomolgus monkey. Treatment with glibenclamide seemed successful for some months, however, the monkey relapsed and was euthanized. Amyloid deposition was present in the islets of Langerhans. Elevated levels of blood glucose, pancreatic enzymes, amylase and lipase, and hypoalbuminemia and protein loss were detected.

### 7.3. Islet Cell Hyperplasia

Islet cell hyperplasia has been reported in New World Monkeys (NWM), baboons, and chimpanzees [9,32,49,50]. It has been linked clinically to disorders of glucose regulation, principally hyperglycemia. The condition is characterized by benign proliferation of pancreatic islet cells [49].

### 7.4. Islet Amyloidosis

Islet amyloidosis has been reported in several NHP [9,51,52,53,54]. Islet amyloid appears to be associated with DM, is difficult to diagnose clinically, and exhibits age-related association [51,52,53,55]. Guardado-Mendoza et al. [9] reported 259 baboons with amyloid deposits in the pancreatic islets, with an increasing incidence with age. Hubbard et al. [52] histologically diagnosed 40 cases of Islet amyloidosis in baboons. Clinical signs such as hyperglycemia and cachexia were sometimes present.

### 7.5. Neoplasia

McClure and Chandler [32] and references therein listed several different pancreatic tumors including islet cell adenoma, ductal adenoma, islet cell adenomatosis, acinar cell adenoma and an adenocarcinoma in Old World Monkeys.

Dias et al. [5] reported a case of an islet cell adenoma in a mantled howler monkey (*Alouatta palliata*) (see Section 5: parathyroid gland).

Hobson and Turner [56] reported a case of a 12-year-old male colobus monkey (*Colobus guereza kikuyuensis*) with a slowly progressive ataxia and paresis which resulted in an acute episode of recumbency, depression, and seizures. Necropsy revealed an 8 cm diameter mass, adherent to the serosa of the proximal duodenum and colon, and embedded within the pancreas and mesenteric root. Histologically, the mass had characteristics of a neuroendocrine or endocrine tumor. Immunohistochemistry confirmed the diagnosis of a mixed pancreatic islet cell tumor.

Simmons and Mattis [11] reported one islet cell adenoma and one case of an exocrine pancreatic adenocarcinoma in rhesus monkeys.

Owsten et al. [57] reported 21 pancreatic neuroendocrine tumors in twelve female baboons. Histologically, all tumors were benign and had the ability to express multiple hormones.

Confer et al. [10] reported seven adenomas and Guardado-Mendoza et al. [9] reported twenty pancreatic islet cell adenomas and one pancreatic islet cell carcinoma in baboons.

## 8. Adrenal Glands

### 8.1. Hyperadrenocorticism

Though common in other species, hyperadrenocorticism (Cushing’s disease) is rarely reported in NHP. All reported cases described clinical signs of dermatologic abnormalities.

Wilkinson et al. [58] reported an adult male rhesus macaque in which a pituitary microadenoma/ectopic ACTH-producing tissue was thought to be the cause of the clinical symptoms of sparse hair coat and thinning of the skin, concurrent with DM. Pituitary-dependent Cushing’s syndrome was subsequently diagnosed by dexamethasone testing, magnetic resonance imaging, and computed tomography (CT) scans. Selegiline therapy, aimed at correcting hypothalamic–pituitary–adrenal axis dysregulation, was not successful; ketoconazole was successful in controlling hypercortisolism.

Jurczynski et al. [59] and Gruber-Dujardin et al. [60] reported a case of an adult female putty-nosed monkey (*Cercopithecus nictitans*) with anamneses of infertility and hyperglucocorticism. Signs of Cushing’s syndrome, evidence for DM, and immunosuppression were evident. Necropsy revealed an adrenal mass (oncocytic adrenocortical carcinoma) with concurrent atrophy of the contralateral gland.

Stoneburg et al. [12] reported a female golden lion tamarin which was evaluated for progressive, bilaterally symmetric alopecia with loss and atrophy of hair follicles and weight loss. Necropsy revealed a pituitary pars intermedia adenoma (see Section 3: pituitary gland) and adrenal cortical hyperplasia consistent with Cushing’s disease. Treatment with ketoconazole was not successful.

### 8.2. Neoplasia

Several different spontaneous adrenal tumors occurring in nonhuman primates are reported in the literature (Table 2).

Beniashvili [3] and references therein indicate that cortical adenomas are frequent in howler monkeys. Also reported were adenoma, carcinoma and adrenal paraganglioma in macaques, and an adrenal pheochromocytoma in a lemur.

In 1996, Dias et al. [5] reported on different cases affecting the adrenal glands of various NWM. The first case was an adult female black-tailed marmoset with generalized weakness and wasting. Gross necropsy revealed marked emaciation, diffuse muscle wasting, alopecia, and multiple skeletal abnormalities. The left adrenal gland was enlarged showing a 3 mm, round pale mass within the cortex on necropsy. It was diagnosed as an actively secreting cortical adrenal adenoma. They also reported two cases of pheochromocytoma in male golden lion tamarins. The first tamarin was euthanized for chronic renal failure and neurologic signs. The left adrenal was enlarged and had a 0.5 mm hemorrhagic friable mass with clusters of neoplastic cells compressing the adjacent cortex and perforating the capsule. The second tamarin died acutely. Pathological findings included acute cholecystitis, chronic nephropathy and cardiomyopathy, and mural tromboendocarditis. Histology revealed a small pheochromocytoma histologically similar to the previous tamarin. The authors go on to describe a case of pheochromocytoma and ganglioneuroma in another male golden lion tamarin. On physical examination, this tamarin was thin and depressed, and a mass was palpated in the cranial ventral abdomen. Chronic cholelithiasis resulting in cholangitis and atrophic pancreatitis was considered the cause of death. A 9 mm multilobulated, firm mass was found on the left adrenal. The mass was composed of two distinctive neoplastic cell populations, indicative of pheochromocytoma and ganglioneuroma. The author also described a case of pheochromocytoma in a mantled howler monkey (see Section 5: parathyroid gland). Finally, they described a case of an adrenal pheochromocytoma and corticotropic cell pituitary adenoma in a 12-year-old male brown spider monkey (*Ateles hybridus*) with long-term cardiomyopathy. At necropsy, classical features of right-sided heart failure were seen. Adrenal glands were bilaterally enlarged with the medulla markedly expanded. Multiple firm nodules, up to 0.5 cm in diameter were found associated with areas of hemorrhage. The cortices were diffusely atrophied. Histologically, the adrenal tumor was diagnosed as a multifocal pheochromocytoma.

Brack [127] reported two adrenal gland tumors in cotton-top tamarins (*Saguinus edipus*): one pheochromocytoma and one cortical adenoma; both associated with morphological signs of myocardial damage and circulatory problems.

In a review of neoplasia of baboons, Cianciolo and Hubbard [7] reported a lipoma of the adrenal gland of a baboon.

Juan-Salles et al. [128] reported benign adrenal neoplasia cases in six NWM, which were diagnosed with unilateral or bilateral adrenal or extra-adrenal pheochromocytoma. Overt invasive behavior or metastases were not evident. All monkeys had myocardial fibrosis, and some had atherosclerosis.

Guardado-Mendoza et al. [9] described adrenal hyperplasia, pheochromocytoma, adenoma, and adrenal carcinoma in baboons. Most animals with pheochromocytomas had clinical signs of hypertension; some showed clinical signs of pulmonary congestion, and one had a seizure as the cause of death. There seemed to be a prediction for adrenal hyperplasia in young overweight female baboons with adrenal hyperplasia. In the same colony, Confer et al. [10] reported two cases of adenomas and three pheochromocytomas in the adrenal glands.

Simmons and Mattis [11] reported sixteen cases of adenomas, one myxoma, one hemangioma, one epithelioid leiomyoma, and one metastatic carcinoma in rhesus monkeys.

### 8.3. Miscellaneous

Weber and Greef [54] reviewed more than 3000 necropsy reports of Chacma baboons (*Papio ursinus*). The most frequent and important finding was that in the majority of baboons with necrotizing cardiomyopathy, adrenal cortical necrosis was also found. Nodular cortical adrenal hyperplasia was observed in approximately 0.5% of all necropsies. The nodules were well circumscribed and similar to small cortical adrenal adenomata in humans.

## 9. Thyroid

### 9.1. Thyroiditis

A survey conducted on 494 Callitrichids revealed 40 cases (8.1%) of histologically evidenced chronic thyroiditis [61]. The authors suggested a genetic component for the occurrence and noted that females were predominantly affected. In one silvery marmoset (*Callithrix argentata*), the pathohistological changes in the thyroid mimicked those of Hashimoto’s disease. In a colony of marmosets. Chalmers et al. [62] found only one case of chronic thyroiditis in 335 examined necropsies.

A case of chronic lymphocytic thyroiditis characterized by multifocal follicular lymphoid cell infiltrates with germinal centers, thyroid acinar atrophy, and pituitary cell hyperplasia/hypertrophy of the adenohypophysis (see Section 3: pituitary gland) was reported in a female cynomolgus monkey [13].

A case of a Hashimoto-like chronic lymphoplasmacytic thyroiditis in an African green monkey was presented by the Armed Forces Institute of Pathology [63].

Guardado-Mendoza et al. [9] reported five cases of thyroiditis in baboons.

Plesker and Hintereder [64] reported a case of an adult female rhesus monkey with no clinical signs suggestive of thyroid abnormality. Pathology revealed a slightly enlarged thyroid gland, with multiple ≤ 2 mm beige-colored foci on the surface of both lobes, histopathologically resembling human Hashimoto’s thyroiditis.

A few other necropsy surveys hint at the possible incidence of chronic thyroiditis in NHP but those animals were enrolled in toxicity studies [65,66].

### 9.2. Hypo- and Hyperthyroidism

Two geriatric rhesus monkeys were described with hyperthyroidism due to multinodular goiter [70]. Clinical signs included an insatiable appetite, hyperactivity, and accentuated ratchet movement and hand tremors while performing fine motor tasks. Bilaterally enlarged thyroid glands could be palpated. Hypertrophic cardiomyopathy was diagnosed in one animal. In both animals, the T3 and T4 levels were elevated. A thyroid biopsy revealed a typical multinodular goiter with cystic hyperplasia. Therapy with thiamazole successfully decreased T3 and T4 levels and diminished hyperactivity but the hypertrophic cardiomyopathy did not resolve.

Lair et al. [67] described a case of a female western lowland gorilla (*Gorilla gorilla gorilla*) having clinical signs of weight gain, unsettled appetite, anxious behavior, lethargy, and showing poor intraspecies interactions. Diagnosis of hypothyroidism was made based on markedly elevated TSH levels and decreased T4 and free T4 levels. Levothyroxine sodium therapy decreased circulating TSH level and significantly improved the animal’s physiologic status and activity level.

McLachlan et al. [68] reported five reports of non-congenital thyroid dysfunction in great apes, four with hypothyroidism (hypothyroidism in a gorilla, [67]) and one with hyperthyroidism.

Fayette et al. [69] reported two cases of congenital hypothyroidism due to thyroid dysgenesis in a nine-month-old male Bornean orangutan (*Pongo pygmaeus*) and a six-week-old female Sumatran orangutan (*Pongo abelii*). The Bornean orangutan was evaluated for delayed growth and development. Genetic testing ruled out genetic disorders. A diagnosis of thyroid dysgenesis was made based on the lack of thyroid tissue on ultrasound exam and radionucleotide uptake on scintigraphy study. The Sumatran orangutan was evaluated for lethargy, poor development, and impaired thermoregulation. No details were provided about the presence or appearance of the thyroid gland in this animal. Although commercial chemiluminescent immunoassays are not validated for use in orangutans, in comparison to age-matched controls, thyroid-stimulating hormone level was markedly elevated, and serum thyroxine (T4) and free T4 levels were markedly decreased in both cases. Oral supplementation with levothyroxine sodium resulted in noticeable clinical improvement in both orangutans within 30 days of initiating treatment [69].

### 9.3. Neoplasia

Tumors of the thyroid gland were observed mainly in macaques and were represented by both adenomas and adenocarcinomas [3]. However, Weber and Greef [54] reported four adenomas of the thyroid of Chacma baboons. All, except one, were microscopic.

Dias et al. [5] reported a case of an adult female black-tailed marmoset presenting with unusual behavior and a protruding upper left canine tooth. At necropsy, the endocrine glands were unremarkable. On histology, the thyroid gland had a sharply demarcated and encapsulated papillary proliferation lined by well-differentiated cuboidal cells within a cystic formation filled with proteinaceous fluid. The neoplasm was identified as a thyroid papillary cystadenoma. Dias et al. [5] also described a case of a 12-year-old male black-tailed marmoset with an acute diarrhea; it died 36 h later due to a diffuse, severe lymphoplasmacytic gastroenteritis. Necropsy revealed the endocrine glands to be grossly unremarkable. However, the thyroid gland presented multiple cystic formations surrounding papillary proliferations lined by well-differentiated cuboidal to cylindrical epithelial cells, compatible with thyroid papillary cystadenoma.

Guardado-Mendoza et al. [9] described in baboons eleven cases of adenomas, eight carcinomas, and two cases of goiter.

In the thyroid of rhesus monkeys seven adenomas, one papillary carcinoma, and one c-cell carcinoma were reported by Simmons and Mattisson [11] and references therein.

Confer et al. [10] described five adenomas in the thyroid gland of baboons and two thyroid follicular carcinomas.

## 10. Reproductive Organs

### 10.1. Ovary

Our search discovered no reports of ovarian pathology in NWM but found that Old World Monkeys are affected by a variety of conditions, notably cystic disease, and granulosa cell tumors [9]. Granulosa cell tumors are tumors of germ cell origin and often result in high estrogen levels. Several review articles [3,11] reported the incidence of ovarian neoplasms including tumor a mucinous cystadenoma, endometrioid tumors, and papillary adenocarcinoma as well as granulosa cell tumors. In a review of spontaneous tumors of ovarian origin, [71] investigated ovarian tumors in a colony of baboons. The authors found that mature and geriatric animals were most affected. Granulosa cell tumors were the most common neoplasm, followed by teratomas and endometrioid carcinomas, with the rare seromucinous cystadenofibroma, cystic papillary adenocarcinoma, and ovarian carcinoma. Weber and Greef [54] reported one a granulosa-theca cell tumor in a chacma baboon, very similar in structure to these well-known tumors in human material.

Other malignant tumors of the ovary have been found in NHP. Ovarian adenocarcinoma has been reported in Bonnet Macaque (*Macaca radiata*) [72,73]. Two cases of ovarian choriocarcinomas are described in macaques. Farman et al. [74] reported a case of a primary ovarian neoplasm of germ cell origin in a clinically healthy adult female rhesus monkey. The monkey had a mass in the right ovary with metastases to the adjacent mesentery and lungs. Neoplastic cell types included cytotrophoblast, syncytiotrophoblast, and extravillous trophoblast. Toyosawa et al. [75] reported an ovarian choriocarcinoma in an adult cynomolgus monkey. Additionally, a mature teratoma in the contralateral ovary was observed. Histology revealed that the choriocarcinoma was characterized by nests of cells where cytotrophoblasts occupied the periphery with syncytiotrophoblasts at the center. The teratoma consisted of well-differentiated epidermal cells, sebaceous glands, hair follicles, cartilage, bone, and teeth. Choriocarcinoma metastasis was observed in multiple organs. Giusti et al. [76] reported a case of an intermediate trophoblast occurring in the ovary of an adult female cynomolgus monkey and Marbaix et al. [77] reported a non-gestational malignant placental site trophoblastic tumor of the ovary in a clinically healthy adult rhesus monkey. In the case of the rhesus, during a laparotomy, a large multinodular mass was discovered by an incident at the site of the right ovary together with several nodules on the adjacent peritoneal area and filled the pelvis. Multiple metastases were observed within the lungs and liver. The tumor was histologically identified as predominantly composed of intermediate trophoblastic cells.

Scott et al. [78] reported an adult female rhesus monkey with a noticeably enlarged right ovary. The histologic evidence of skin within ovarian tissue led to classification as a teratoma. On an experimental laparotomy of a healthy adult pigtailed macaque (*Macaca nemestrina*) of proven fertility, large firm dark ovaries were noted [73]. Although both ovaries exhibited abnormalities, an ovarian teratoma was observed on the right only. Mature tissues of ectodermal, endodermal, and mesodermal origin, including squamous epithelium, glial tissue, adipose tissue, intestinal tissue, and cartilage were detected. Baskin et al. [79] reported an African Green Monkey (*Cercopithecus aethiops*) with a benign cystic teratoma of the ovary. Abdominal distension of a non-pregnant animal was the only notable clinical sign. The tumor was predominantly of endodermal origin.

Amin et al. [80] described the ultrastructural characteristics of a surface papilloma and a serous cystadenofibroma which were observed by accident in the ovary of a normal menstruating rhesus monkey. DiGiacomo [81] reported a serous cystadenoma in a female rhesus monkey. Graham and McClure [82] reported ovarian tumors (i.e., fibrothecomas and a Sertoli-Leydig cell tumor) and related lesions in aged chimpanzees. Holmberg et al. [83] reported a dysgerminoma in a female rhesus monkey that was euthanized because of additional metastatic lesions after the initial surgical removal of an ovarian tumor. An ovarian mucinous cystadenoma was reported in an adult female cynomolgus monkey [84]. The tumor was constituted of various sizes of multilocular cystic glands lined by a single layer of mucin-filled epithelium.

Marr-Belvin et al. [85] described a retrospective analysis of 458 female rhesus monkey necropsies. Degenerative and inflammatory changes in the ovaries included mineralization, infiltration by lymphocytes, macrophages and multinucleated giant cells, endometriosis, and arteriopathy. Cystic changes included follicular cysts, cystic rete, and mesonephric duct cysts with cystic rete the most common. Neoplasms included granulosa cell tumors, cystadenoma, cystadenocarcinoma, and teratoma. Weber and Greef [54] reviewed 3000 necropsies of chacma baboons. They found ovarian cysts in 1.5% of the cases. All were non-neoplastic and represented follicle cysts, germinal epithelial inclusion cysts, and cysts of parovarian origin.

### 10.2. Testis

Isolated cases of tumors of the testicles are reported. While examining 314 howler monkeys (*Aloutta caraya*), Maruffo and Maalinow [86] found one with a testicular tumor. The testis was slightly enlarged with an 8 mm nodule that was diagnosed as a seminoma. No clinical signs were associated with this finding. Jones et al. [87] reported a case of an interstitial cell tumor of the testis in an adult male Western Lowland Gorilla. The postmortem exam revealed very small testes, and the cut surface of the left testis appeared nodular. Two interstitial cell adenomas were diagnosed in the left testis. No clinical signs were associated with this finding. Miller and Boever [88] reported a case of a Sertoli cell tumor in a black lemur (*Lemur macaco*). The animal presented with an acute and apparently painful onset of swelling of the right inguinal region. Histological examination revealed a Sertoli cell tumor. Gozalo et al. [89] reported a case of an adult Owl monkey (*Aotus nancymae*), a proven breeder, who was submitted for surgery due to a suddenly markedly swollen testicle. Microscopically, the morphology of the tumor cells was consistent with a seminoma.

Androgenetic alopecia is described in stump-tailed macaques and is so-called ‘male-pattern baldness’. It entails hereditary balding similar in many respects to that of androgenetic alopecia in humans [129]. Increasing levels of testosterone, occurring at approximately four years of age, are associated with typical signs of male-pattern baldness along the frontal scalp.

## 11. Summary and Conclusions

This is the first systematic review of naturally occurring endocrinologic disorders in NHP. NHP can become ill due to a variety of endocrine disorders. However, in most cases, no clinical signs were noted and on gross pathology, the endocrine organs were unremarkable. The diagnosis was frequently made as incidental findings after standard histological examination. While in most of the cases, diagnoses were made based on incidental findings at necropsy, with no apparent premortem disease, we feel that a thorough review of endocrine disorders in NHP is a worthwhile endeavor. Furthermore, these findings were frequently incidental yet have the potential to impact studies. This review makes it evident that standard procedures for diagnosing, monitoring, or treating endocrine disorders in NHP are lacking. Further research should be done to evaluate these procedures and establish risk factors.

## Figures and Tables

**Table 1 animals-12-00407-t001:** Spontaneous endocrine disorder in nonhuman primates: distribution by gland.

Chapter	Gland	Endocrine Disorder	Reference
2	Hypothalamus	None	
3	Pituitary gland	Neoplasia	[2,3,4,5,6,7,8,9,10,11,12]
		Hyperplasia	[10,13]
4	Pineal gland	None	
5.1	Parathyroid gland	Hyperparathyroidism	[5,9,14,15,16]
5.2		Neoplasia	[9,10,11]
6	Thymus	Neoplasia	[17]
7.1	Pancreas	Diabetes mellitus	[18,19,20,21,22,23,24,25,26,27,28,29,30,31,32,33,34,35,36,37,38,39,40,41,42,43]
7.2		Acute and chronic pancreatitis	[32,44,45,46,47,48]
7.3		Islet cell hyperplasia	[9,32,49,50]
7.4		Islet amyloidosis	[9,51,52,53,54,55]
7.5		Neoplasia	[5,9,10,11,32,56,57]
8.1	Adrenal glands	Hyperadrenocorticism (Cushing’s disease)	[12,58,59,60]
8.2		Neoplasia	Table 2
8.3		Miscellaneous	[54]
9.1	Thyroid	Thyroiditis	[9,13,61,62,63,64,65,66]
9.2		Hypothyroidism	[67,68,69]
9.2		Hyperthyroidism	[68,70]
9.3		Neoplasia	[3,5,9,10,11,54]
10.1	Reproductive organs	Ovary neoplasia	[3,9,11,54,71,72,73,74,75,76,77,78,79,80,81,82,83,84,85,86,87,88]
10.2		Testes neoplasia	[86,87,88,89]

**Table 2 animals-12-00407-t002:** Distribution of spontaneous adrenal neoplasms in nonhuman primates, by species.

Histologic Diagnosis	Species	Reference
Adenoma	Howler monkey	[3]
	Cotton top tamarin	[127]
	Black-tailed marmoset	[5]
	Macaque	[3]
	Baboon	[9,10]
Carcinoma	Macaque	[3]
	Baboon	[9]
Paraganglioma	Macaque	[3]
Pheochromocytoma	Golden lion tamarin	[5]
	Cotton top tamarin	[127]
	Mantled howler monkey	[5]
	Lemur	[3]
	Black howler monkey	[128]
	Black handed spider monkey	[128]
	Baboon	[9,10]
Ganglioneuroma	Golden lion tamarin	[5]
Lipoma	Baboon	[7]
Hyperplasia	Baboon	[9]

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
