# Peer review of "Naturally Occurring Endocrine Disorders in Non-Human Primates: A Comprehensive Review"

_animals, 2022, doi:10.3390/ani12040407_

Round 1

Reviewer 1 Report

This is a well written review of the literature regarding naturally occurring endocrine disorders.  The work is interesting and may benefit some researchers in the field.  The primary weakness in the writeup is that there is no presentation of the search strategies used to assure that disorders have not been missed in the writeup.  For instance, mammary hyperplasia is listed (a target of estrogen/progesterone action) but no mention of endometriosis (also a target of estrogen action).  Without a clear description of how the search was  conducted (with parameters) the impact of the review is less helpful.  They also need to show a breakdown of how many hits they had on their search criteria.  In addition,  a significant amount of older work on nonhuman primates was published in book form.  Did the search  also address older work or work that is published in book form.

Author Response

Dear reviewer,

We would like to thank you for your comments and remarks. We have revised the manuscript according to your remarks. Please find below our point-to-point reply.

As suggested by the reviewer, a careful proof read was performed and the English language and style errors were corrected.

We fully agree with your comments. Due the complexity of the literature search, we decided not to write and submit this review as a ‘systematic review’ (no Prisma checklist and Prospero registration needed) but as a regular review. To clarify, we have added in the introduction:

To these ends, we first did a literature search for books, book chapters, peer-reviewed publications, conference proceedings, and newsletters in academic literature databases, such as Google Scholar, PubMed, BioOne Complete, and Web of Science, using words and word combinations, such as endocrine disease, thyroid, pancreas, diabetes, monkeys. We then evaluated the search results for those reports which we considered as clinically relevant and then divided them into the endocrine organs (Table 1).

We agree with the comments that we should focus on the primary endocrine diseases and not the target organs so we removed the mammary glands paragraph. Thank you for pointing this out to us.

Kind regards,

Jaco Bakker

Reviewer 2 Report

I read with interest this review. Veterinary endocrinology being my area of expertise, I was also very interested in reviewing this paper. At the same time, I was worried and intrigued by the Specie (NHP) for which I have no knowledge. I was also interested to learn about the contribution of non-human primates (NPH) to endocrinology and to learn more about endocrine disorders. It is agreed that the submitted manuscript is potentially useful and timely. This is the first systematic review of naturally occurring endocrinological disorders in NHP, properly written and with scientific rigor. An accurate synthesis of the various previous studies in NHP was carried out. It represents a good contribution to the field and to existing knowledge. The review is attractive, easy to read, clearly and nicely presented and well organized. It contains all components expected. The sections are well development and there is a fair balance between the different sections. It is technically sound. Abstract and introduction clearly identify the need for this research and its relevance. It explains why the theme is important in terms of the problems to be investigated, the context for the research question, what place this research question has in understanding the topic. The aim set out by the authors in the manuscript is adequately achieved and well discussed. Tables are clear and fully described. Literature is suitable and authors synthesized the current literature properly.  I found no weaknesses or limitations through the whole manuscript. I have no specific comments to suggest for authors in order to improve the quality of manuscript. English language and style are fine. The review contains useful data which deserve to be published. I recommend that this review be accepted for publication in current form.

Author Response

Dear reviewer,

We would like to thank you for your positive comments.

Kind regards,

Jaco Bakker

Reviewer 3 Report

This is a well-written review containing important information for readers.

Only minor editorial revisions will be needed.

  1. Past tense and present tense are used inconsistently, therefore, the authors may need to revise this issue throughout the text.
  2. line 29: Diabetes Mellitus should be Diabetes mellitus
  3. line 46: add commas before and after therefore
  4. lines 150, 152, 153, 244, 271, 281: diabetes mellitus should be DM
  5. line 313: 6.98% should be 7.0%
  6. line 368: add a comma after 1996
  7. line 469: thyroid immunoassays?  thyroid hormone ELISA?
  8. line 544: add Baskin et al. before [127]
  9. line 580: sertoli should be Sertoli
  10. line 584: a should not be italic
  11. References: scientific names of animals should be italic, titles should be small capitals
  12. line 770: underline should be deleted
  13. line 772: journal name should be appropriately abbreviated

Author Response

Dear reviewer,

We would like to thank you for your comments and remarks. We have revised the manuscript according to your remarks where possible. Please find below our point-to-point reply:

  1. As suggested by the reviewer, a careful proof read was performed and the mentioned English language spell errors (past tense and present tense) were corrected throughout the whole manuscript.
  2. We agree and have adapted this.
  3. Adapted as requested.
  4. We agree and adapted this.
  5. We agreed, adapted as requested throughout the whole manuscript (1 decimal behind the comma).
  6. Inserted as requested.
  7. Thank you for bringing this up to us. We have inserted in line 571 ‘chemiluminescent’ to make clear what kind of assay was used.
  8. Done as suggested.
  9. Adapted as suggested.
  10. 10-12: References are inserted by Endnote. Journal abbreviation and underline were corrected.

Kind regards,

Jaco Bakker

Reviewer 4 Report

In this article the authors make a comprehensive review of naturally occurring endocrine disorders in non-human primates. The review seems interesting to me, it compiles all the endocrinological processes documented in the literature to date.

However, there are some points that the authors should take into account when writing their study correctly, which I am sure will provide greater understanding and clarity to the review.

Keywords: put in alphabetical order

Introduction:

Line 54: Pituitary, parathyroid, pancreas, adrenal glands, thyroid and reproductive organs include pathologies in the same grid of the table. I would include the references of the various studies in the table. It would be easier to identify the different types of diagnosed injuries. For example, table II faithfully follows the review study, I would apply the same criteria to table I.

Line 58: What kind of effect does age produce?

Lines 91-92: Cianciolo & Hubbard [9] and references therein report a pituitary carcinoma in a baboon?. Not understood, please clarify.

List each section and subsection, for example:

  1. Hypothalamus
  2. Pituitary Gland:

2.1.-Neoplasia

2.2.- hyperplasia

And so on……..

It would be easier to follow the paper.……,

If the authors refer to the past in the studies, they must do so in all of them.

Brack [98] reports or reports………….

Juan-Salles et al. [99] reporto r reported?.............

Lines 411-413: Simmons & Mattis [14] and references therein reported sixteen cases of adenomas, one myxoma, one hemangioma, one epithelioid leiomyoma, and one metastatic carcinoma in rhesus monkeys. Rewrite, it is not understood.

References:

If there is no DOI, include PMID. For example:

  • 5. PMID: 2693732
  • 8. PMID: 11560408

Author Response

Dear reviewer,

We would like to thank you for your comments and remarks. We have revised the manuscript according to your remarks where possible. Please find below our point-to-point reply.

- As suggested by the reviewer, a careful proof read was performed and the mentioned English language spell errors (past tense and present tense) were corrected throughout the whole manuscript.

- The keywords were inserted in alphabetic order.

- We agree with this comment and we have adapted Table I accordingl to Table II.

- We have implemented, as proposed, the chapter numbering.

- In line 58 the age effect is inserted in the manuscript: Basal plasma levels of adrenal androgenes and early precursors of steroid hormones progressively decrease with age, while cortisol concentrations do not change.

- We understand the confusion and adapted the manuscript to “ Cianciolo & Hubbard [9] reported one pituitary carcinoma of in a baboon.” and to “Simmons & Mattis [14] reported one islet cell adenoma and one case of an exocrine pancreatic adenocarcinoma in rhesus monkeys.”

- we have added all missing PMID.

Kind regards,

Jaco Bakker

Round 2

Reviewer 1 Report

All points addressed.